# Establishment of Tibetan-Sheep-Specific SNP Genetic Markers

**Benmeng Liang** [1,2,3,†], **Yuhetian Zhao** [1,3,†], **Yabin Pu** [1,3], **Xiaohong He** [1,3], **Jiangang Han** [1,3,4], **Baima Danzeng** [5], **Yuehui Ma** [1,3], **Jianfeng Liu** [2,*] and **Lin Jiang** [1,3,*]

[1] National Germplasm Center of Domestic Animal Resources, Ministry of Technology, Institute of Animal Sciences, Chinese Academy of Agricultural Sciences (CAAS), Beijing 100193, China
[2] National Engineering Laboratory for Animal Breeding and MOA Key Laboratory of Animal Genetics and Breeding, College of Animal Science and Technology, China Agricultural University, Beijing 100193, China
[3] Key Laboratory of Livestock and Poultry Resources (Cattle) Evaluation and Utilization, Ministry of Agriculture and Rural Affairs, Beijing 100193, China
[4] Animal Genomics Laboratory, UCD School of Agriculture and Food Science, UCD College of Health and Agricultural Sciences, University College Dublin, Belfield, D04 V1W8 Dublin, Ireland
[5] Nagqu City Animal Husbandry and Veterinary Technology Promotion Station, Nagqu 852000, China
*   Correspondence: liujf@cau.edu.cn (J.L.); jianglin@caas.cn (L.J.)
†   These authors contributed equally to this work.

**Abstract:** Tibetan sheep are one of the three major coarse sheep breeds in China, and they possess a long history of formation. However, few studies have been conducted on the identification of Tibetan sheep breeds at the molecular level. In this study, a total of 448 individuals from 24 Tibetan sheep populations in the 5 regions of Tibet, Qinghai, Gansu, Yunnan, and Sichuan were analyzed using the Affymetrix Ovine SNP 600K high-density chip to construct specific single-nucleoside polymorphism (SNP) genetic marker sets of Tibetan sheep breeds. Firstly, the genetic structure analysis showed that Yunnan–Tibetan sheep (NL, Ninglang; JC, Jianchuan), Zuogong (ZG), Heizang (HZ), Gongga (GG,) and Tao sheep (TS) can be clearly distinguished from other Tibetan sheep populations. Next, based on the population differentiation index FST, the PCA and NJ tree results showed that only 60 specific SNPs can successfully separate Tibetan sheep in the Yunnan region from Tibetan sheep in other regions, and the distinguishing effect on Yunnan–Tibetan sheep reached 100%. Using the same method, we found that 4 Tibetan sheep breeds, including Zuogong (ZG, 20 SNPs), Heizang (HZ, 60 SNPs), Gongga (GG, 60 SNPs), and Tao sheep (TS, 30 SNPs), can also be distinguished from other Tibetan sheep populations with only a few SNP loci (20–60), and the distinguishing effect reached 100%. Overall, we successfully obtained a Yunnan region-specific (60 SNPs) genetic marker set and 4 breed-specific SNP genetic marker sets (20–60 SNPs) for the first time for the identification of Tibetan sheep breeds at the molecular level. These made up for the lack of genetic marker sets for the identification of Tibetan sheep breeds, and provided a genomic basis for the scientific classification and accurate identification of livestock and poultry genetic resources on the Qinghai–Tibet Plateau.

**Keywords:** Tibetan sheep; 600K high-density chips; population differentiation index FST; genetic marker sets

## 1. Introduction

Tibetan sheep are one of the three major coarse wool sheep breeds in China and have a long history of formation. Previous studies have shown that Tibetan sheep originated in northern China in the late Holocene, dating back to 1800–4500 years ago [1–3]. Then, along the valley of the Hengduan Mountain Range at the eastern end of the Qinghai–Tibet Plateau, they gradually spread across the Qinghai–Tibet Plateau in high-altitude areas, such as Tibet, Qinghai, Gansu, Yunnan, and Sichuan over generations. Currently, Tibetan sheep are the most numerous livestock (>20 million) on the Qinghai-Tibet Plateau. Additionally, they play an important role in the economy, culture, and society of Tibetan pastoral areas [4–6]. However, due to the introduction of reform measures, such as grassland contracting to

households, private breeding, and the continuous implementation of herdsmen's management autonomy, as well as the economic interests of meat use, the genetic diversity of Tibetan sheep has gradually declined, and the protection of germplasm resources has been severely tested. Therefore, research on the resource evaluation of Tibetan sheep is of practical significance for their genetic breeding, rational protection, and utilization.

Through long-term natural selection, the complexity of the high-altitude environment has created a variety of Tibetan sheep breeds, mainly reflected in their morphology, physiology, and genetics [7,8]. It was difficult to identify and classify these breeds based on their origin, body appearance, and production performance. Additionally, due to the limitations of identification technology, a phenomenon of "fish eyes mixed with pearls" was identified in the market. Many "counterfeit" livestock and poultry breeds were observed that appeared to be good breeds but had poor performance. Compared with past molecular technologies, such as DNA fingerprinting [9], DNA barcoding [10], and SNP barcoding [11], today's high-throughput, high-density SNP markers [12–14] have greatly improved the efficiency of genetic detection and offer the possibility to track the selection experienced by different populations. The previous technology was not only time-consuming, laborious, and costly, but also very complex to apply, and for farms and farmers, it was impossible to bear the cost of breed identification of their own Tibetan sheep. The construction of specific SNP genetic markers can not only distinguish Tibetan sheep breeds with very few SNP sites, but also reduce costs. At the same time, specific SNP genetic markers are simple to operate, only requiring the collection of tissue or blood samples. It is convenient for farms and farmers to identify their own sheep breeds. Therefore, based on re-sequencing and high-density chip detection technology, the classical genetic method of selection signal analysis was used to screen out breed-specific genetic loci, and to identify and differentiate Tibetan sheep breeds germplasm resources at the molecular level.

In recent years, great progress has been made regarding the origin, evolution, genetic structure and genetic diversity of sheep in China, and some studies have also involved Tibetan sheep, but most of these studies have used traditional mtDNA D-loop sequences [6] and microsatellite labeling technology [15] to analyze the genetic structure of Tibetan sheep. With the advent of high-throughput sequencing and chip technology, Menghua Li (2016) et al. [2] used whole-genome resequencing technology to analyze the genetic structure of only 77 individuals of 24 breeds from the Qinghai–Tibet Plateau, the Yunnan–Guizhou Plateau, North China, and East China, and found that there was genetic mixing between the Qinghai–Tibet Plateau and sheep breeds in northern and eastern China, and genetic infiltration of northern and eastern Chinese breeds into Yunnan–Guizhou breeds. In 2019, Li Menghua et al. [16] conducted a population structure analysis of 186 sheep from the Qinghai–Tibet Plateau, the Yunnan–Guizhou Plateau, and northern China, and found that the genetic affinity between Tibetan sheep and the Yunnan–Guizhou population was closer than that between Tibetan sheep and the northern Chinese population (i.e., the ancestral population of Tibetan sheep). However, these studies mainly focused on population structure origin and evolution study, and to date no study has identified specific SNP markers for the identification of Tibetan sheep breeds.

In order to construct specific single-nucleoside polymorphism (SNP) genetic marker sets for the identification of Tibetan sheep breeds at the molecular level, this study enriched the number of Tibetan sheep breeds and individuals; using 600K chip technology, a total of 24 Tibetan sheep groups (448 individuals) in the 5 provinces were subjected to SNP chip detection and in-depth analysis of their population genetic structure and, at the same time, selection signal analysis was used to screen out breed-specific genetic loci. This enabled the identification of the germplasm resources of Tibetan sheep breeds at the molecular level, aiming to combine modern biological information and molecular biological technology with animal husbandry. The germplasm resource identification technology can be applied to breed identification, anti-counterfeiting, breeding, production, management, and other fields of livestock and poultry.

## 2. Materials and Methods

### 2.1. Ethics Statement

All experiments in this study involving animals were conducted according to the ethical policies and procedures approved by the Animal Care and Use Committee of the Chinese Academy of Agricultural Sciences and the Ministry of Agriculture of the People's Republic of China (IAS2019-57).

### 2.2. Sample Collection

In this study, a total of 448 unrelated adult individuals from 24 Tibetan sheep populations in 5 provinces, including 192 samples from 10 Tibet–Tibetan sheep breeds, 90 samples from 5 Qinghai–Tibetan sheep breeds, 75 samples from 4 Gansu–Tibetan sheep breeds, 55 samples from 3 Sichuan–Tibetan sheep breeds, and 36 samples from 2 Yunnan–Tibetan sheep breeds, were investigated (Supplementary Table S1). All experimental sheep were randomly selected. The basic information and phenotypic data of individuals were included in the phenotypic database of livestock and poultry genetic resources on the Qinghai–Tibet Plateau, and all tissue samples were used for subsequent DNA extraction and SNP BeadChip genotyping assay.

### 2.3. SNP BeadChip Genotyping and Quality Control

A total of 448 samples were genotyped using Affymetrix Ovine 600K BeadChip (Beijing Compass Agritechnology Co., Ltd., Beijing, China). A Promega genomic DNA extraction kit was used to extract genomic DNA. A total of 482,709 SNPs were generated (based on genome Oar_v3.1). The obtained data was used for quality control with PLINK (Version 1.90 b) software, and unqualified individuals and SNPs were eliminated, and the quality control standards were as follows: (1) call rate $\geq 90\%$; (2) minor allele frequency (MAF) $\geq 0.01$; (3) Hardy–Weinberg equilibrium (p) $< 1 \times 10^{-5}$. In order to study the genetic structure between samples and avoid the impact of interlinked SNPs on the analysis of population structure, after PLINK trimming, SNPs whose linkage disequilibrium coefficient (r2) was less than 0.5 were used to construct population structure analysis, mainly including PCA and NJ genetic evolution tree analysis. Finally, 436 individuals and 373,991 common SNPs were used to establish a population differentiation index and genetic marker sets.

### 2.4. Population Structure, Phylogenetic Analysis

All 373,991 common SNPs were pruned using PLINK [17] by the parameter (–indep-pairwise 1000 5 0.5). This obtained 238,405 independent SNPs for subsequent analyses of population structure analysis. The PCA and NJ tree analysis were conducted using the PLINK software. The Interactive Tree Of Life (ITOL) online tool was used to visualize the phylogenetic trees (https://itol.embl.de, accessed on 2 July 2021) [18].

### 2.5. Calculation of Population Differentiation Index

The FST values, named as the fixation index, are typically used to evaluate differentiated genomic regions and identify selective signals among whole-genomic sequences. The population differentiation index FST is an important indicator to detect whether there is genetic differentiation between populations, and the degree of differentiation between populations is reflected through the difference in allele or genotype frequency between populations [19]. Population differentiation values (FST fixation) [20] for each population were calculated basing on a sliding window size of 10 kb and a step size of 1 kb. Then, the detection populations were divided into a specific experimental group and a control group, and the vcftools_0.1.13 software [21] was used to calculate the FST value of each SNP marker locus in the whole genome (the value range of the FST statistic is between 0–1; the larger the number, the greater the separation degree of the population at the observation site), and the candidate-specific SNP sites of each Tibetan sheep population were screened out according to their FST threshold value. Then, the number of candidate-specific SNP sites were decreased sequentially in the order of FST value (from small to large), further

detecting the discriminative effect of different numbers of SNPs on Tibetan sheep populations using the PCA and NJ tree and confirming a high-quality region-specific locus of Tibetan sheep.

## 3. Results

### 3.1. Genomic Variation and Population Structure Analysis

A total of 448 individual Tibetan sheep from 24 geographic regions of China (Figure 1A and Supplementary Table S1) were recruited for Affmetrix Ovine 600k SNP array analysis, including 192 samples from 10 Tibet–Tibetan sheep breeds, 90 samples from 5 Qinghai–Tibetan sheep breeds, 75 samples from 4 Gansu–Tibetan sheep breeds, 55 samples from 3 Sichuan–Tibetan sheep breeds, and 36 samples from 2 Tibetan sheep breeds. After stringent quality filtering, a final set of 436 sheep individuals and 373,991 common SNPs were retained in the downstream population structure analysis and SNP genetic marker set establishment. The PCA based on the first two components (explaining 14.97% of total variation) showed that PC1 can explain 8.64% of the genetic variation, and clearly distinguished Yunnan–Tibetan sheep (NL, Ninglang; JC, Jianchuan) and Sichuan Gongga (GG) sheep from all groups; PC2 can explain 6.33% of the genetic variation, and we found the Tibet Zuogong sheep (ZG) clustered together alone, showing a higher degree of differentiation (Figure 1B). In addition, the Qinghai Black Tibetan sheep (HZ) and Tibet-valley-type Tibetan sheep (SG) were also can be distinguished from other Tibetan sheep groups (Figure 1B). This showed that they have a higher degree of differentiation trend. Except for these 6 breeds, further PCA among the remaining 18 Tibetan sheep populations showed that Huoerba sheep (HB), Jiangzi sheep (JZ), Langkazi sheep (LKZ), Shangu sheep (ValTib), grassland sheep (PraTib), grassland sheep (GraTib), and Gangba sheep (GB) from Tibet can be distinguished from Tibetan sheep in other regions (Figure 1C), while Gansu Tao sheep also clustered alone and were indeed separated from 18 sheep breeds, showing they had an independent genetic background. Finally, Duoma (DM) and Awang sheep (AW) were relatively isolated, and the other eight Tibetan sheep populations had a higher genetic structure similarity, which was difficult to further distinguish (Figure 1D,E). Therefore, these eight Tibetan sheep groups, including Gansu Ganjia sheep (GJ), Gansu Oula sheep (GOL), Gansu Qiaoke sheep (QK), Qinghai Oula sheep (QOL), Sichuan Jialuo sheep (JL), Qinghai Qilian Tibetan sheep (QL), Qinghai Tibetan sheep (QT), and Zhaxika sheep (ZX), were regarded as a mixed population for the following genetic differentiation index (FST) calculation. Subsequently, phylogenetic analysis using the distance-based NJ tree also supported these results (Figure 2).

### 3.2. Population Genetic Differentiation Distance (FST)

Based on a population structure analysis of Tibetan sheep, 24 populations were preliminarily divided into 17 groups, including 16 Tibetan sheep and a mixed Tibetan sheep population. However, the population structure analysis results could not quantitatively determine the degree of genetic differentiation between these groups, so this study used SNP data based on a 600K high-density chip to accurately calculate the genetic differentiation index (FST) at the genome level. For an average genetic differentiation index (FST) greater than 0.05, according to the criteria proposed in the literature, it is considered that the population has a medium degree of genetic differentiation.

Population genetic differentiation distance (FST) results found that the average genetic differentiation index FST of the population was lower than 0.04 (FST = 0.03) between the mixed populations and the other 16 populations; however, the average genetic differentiation index (FST) among the other 16 populations was greater than 0.04. Among them, the FST values of Tibet Zuogong sheep (ZG, 0.08), Qinghai Black Tibetan sheep (HZ, 0.07), Yunnan Ninglang sheep (NL, 0.07), Yunnan Jianchuan sheep (JC, 0.07), Sichuan Gongga sheep (GG, 0.06), Tibet Jiangzi sheep (JZ, 0.06), Qinghai Valley Tibetan sheep (SG, 0.06), Tibet Valley Tibetan sheep (ValTib, 0.06), Tibet Gangba sheep (GB, 0.05), Tibet Gaize grassland Tibetan sheep (GraTib, 0.05), and Gansu Tao sheep (TS, 0.05) were all between

0.05–0.1, and there was moderate differentiation between the groups, while the FST value of Huoerba sheep (HB, 0.04), Tibetan Awang sheep (AW, 0.04), Tibetan Duoma sheep DM (DM, 0.04), Tibetan Langkazi (LKZ, 0.04), and Tibet Dangxiong grassland Tibetan sheep (PraTib, 0.04) ranged from 0.04 to 0.05, and there was also a certain degree of differentiation between groups (Figure 3). The results were consistent with the PCA. This provided a genomic basis for the scientific classification and accurate identification of Tibetan sheep genetic resources.

### 3.3. Construction of Region-Specific SNP Marker Sets

The Tibetan sheep population that reached the degree of moderate differentiation between breeds was screened for specific SNP marker sets. Then, based on the provinces and ecological types of Tibetan sheep, four breeds from different regions were screened and used to construct breed-specific SNP marker sets. Tibet Zuogong sheep (ZG) (0.08), Qinghai Black Tibetan sheep (HZ) (0.07), Sichuan Gongga sheep (GG) (0.06), Gansu Tao sheep (TS) (0.05), and two breeds from the Yunnan provinces were screened and constructed for Yunnan-region-specific SNP marker sets, including Yunnan Ninglang sheep (NL) (0.07) and Yunnan Jianchuan sheep (JC) (0.07).

Firstly, taking Tibetan sheep in Yunnan (Ninglang sheep, NL) (0.07) and Yunnan Jianchuan sheep, JC) (0.07) as the test population and all Tibetan sheep in other regions as the control population, the allele frequency differences between Tibetan sheep in Yunnan and Tibetan sheep in other regions were calculated based on the population differentiation index (FST). A total of 176 candidate-specific SNPs with an FST value greater than 0.5 were screened (Figure 4A). After pruning, 126 SNPs remained. The number of candidate-specific SNPs was decreased in the order of FST value (from small to large), and finally the differential efficacy of different numbers of SNPs was detected by using a PCA and NJ tree on Yunnan–Tibetan sheep. The PCA results showed that the differentiation effect of these 126 SNPs was remarkable, and PC1 clearly distinguished Yunnan–Tibetan sheep (Ninglang sheep and Jianchuan sheep) from all Tibetan sheep populations (Figure 4B). In line with the PCA results, the NJ tree also showed that Yunnan–Tibetan sheep individuals clustered together to form a single group (Figure 4C). By continuously reducing the number of candidate-specific SNPs to 60, it can be seen that 60 SNPs can still distinguish Tibetan sheep from the herd; although the PCA results showed that the individuals of Tibetan sheep in Yunnan were more dispersed, the discrimination power can still reach 100% (Figure 4D). The distinguishing effect of the PCA and NJ tree on Tibetan sheep in Yunnan was basically consistent (Figure 4E), so the number of the least and most accurate specific SNP sites that can be used as a specific genetic marker set for Yunnan sheep is about 60 SNPs (Supplementary Table S6).

### 3.4. Construction of Breed-Specific SNP Marker Sets

Among these Tibetan sheep, Tibetan Zuogong sheep (ZG) had the highest level of differentiation. Tibetan Zuogong sheep (ZG) were then used as the experimental population and the other Tibetan sheep were used as control groups. Then, according to the population differentiation index, we calculated the allele frequency differences between Zuogong sheep and other Tibetan sheep at each marker site. Finally, 107 candidate specific SNP sites with FST > 0.65 were screened (Figure 5A). After pruning, 90 SNPs were used for subsequent PCA- and NJ-tree-distinguishing efficacy detection. Both PCA and NJ tree methods could clearly distinguish the Zuogong sheep from the other Tibetan sheep groups to gather chaotically in the group. The PCA results showed that the Zuogong sheep population were scattered but clustered separately (Figure 5B). The NJ tree results also showed that Zuogong sheep individuals were clustered individually and had independent genetic backgrounds (Figure 5C). When the number of SNPs was reduced to 20, the Zuogong sheep could still be distinguished from the Tibetan sheep population (Figure 5D,E). Therefore, these 20 SNPs could be used as the specific genetic tag of the Zuogong sheep.

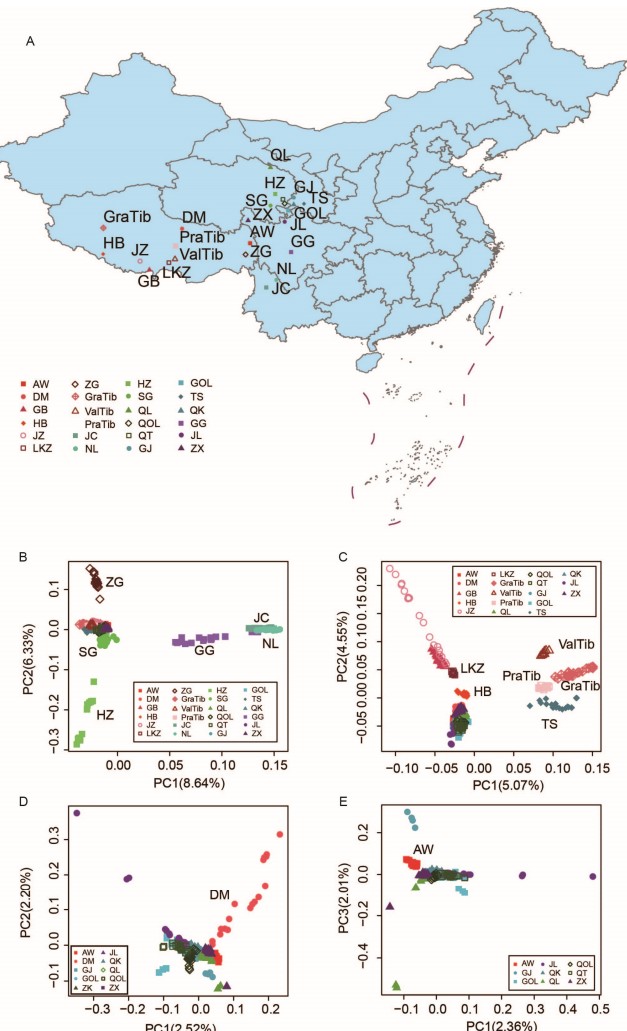

**Figure 1.** (**A**) A map of the studied area along with the distribution of the 24 sheep populations. (**B**) PCA plot of the first 2 components of 24 sheep breeds. (**C**) PCA plot of the first 2 components of 18 sheep breeds. (**D**) PCA plot of the first 2 components of 10 sheep breeds. (**E**) PCA plot of the first and the third component of nine sheep breeds. The fraction of the total variance explained is reported on each individual axis between parentheses. The red color represents Tibetan sheep in Tibet (N = 180—AW, Awang sheep from Gongjue County in Tibet, 19; DM, Duoma sheep from Anduo County in Tibet, 19; GB, Gangba sheep from Gangba County in Tibet, 37; HB, Huoerba sheep from Zhongba County in Tibet, 19; JZ, Jiangzi sheep from Jiangzi County in Tibet, 18; LKZ, Langkazi sheep from Langkazi County in Tibet, 9; ZG, Zuogong sheep from Zuogong County in Tibet, 18; ValTib, Shangu sheep from Gongga County in Tibet, 8; PraTib, grassland sheep from Dangxiong County in Tibet, 10; GraTib, grassland sheep from Gaize County in Tibet, 23). The cyan color represents Tibetan sheep in Yunnan (N = 36—JC, Jianchuan sheep from Jianchuan County in Yunnan, 18; NL, Ninglang sheep from Ninglang Autonomous County in Yunnan, 18). The green color represents Tibetan sheep in Qinghai (N = 90—HZ, Heizang sheep from Guinan County in Qinghai, 18; SG, Shangu sheep from Maqin County in Qinghai, 18; QL, Qilian sheep from Qilian County in Qinghai, 18; QT, Qinghai Tibetan sheep from Zeku County in Qinghai, 18; QOL, Oula sheep from Henan County in Qinghai, 18). The blue color represents Tibetan sheep in Gansu (N = 75: GOL, Oula sheep from Maqu County in Gansu, 18; GJ, Ganjia sheep from Xiahe County in Gansu, 18; QK, Qiaoke sheep from Luqu County in Gansu, 18; TS, Tao sheep from Zhuoni County in Gansu, 21). The purple color represents Tibetan sheep in Sichuan (N = 55—GG, Gongga sheep from Luding County in Sichuan, 18; JL, Jialuo sheep from Aba County in Sichuan, 18; ZXK, Zhaxika sheep from Shiqu County in Sichuan, 19). The shapes of the symbols indicate the geographic regions.

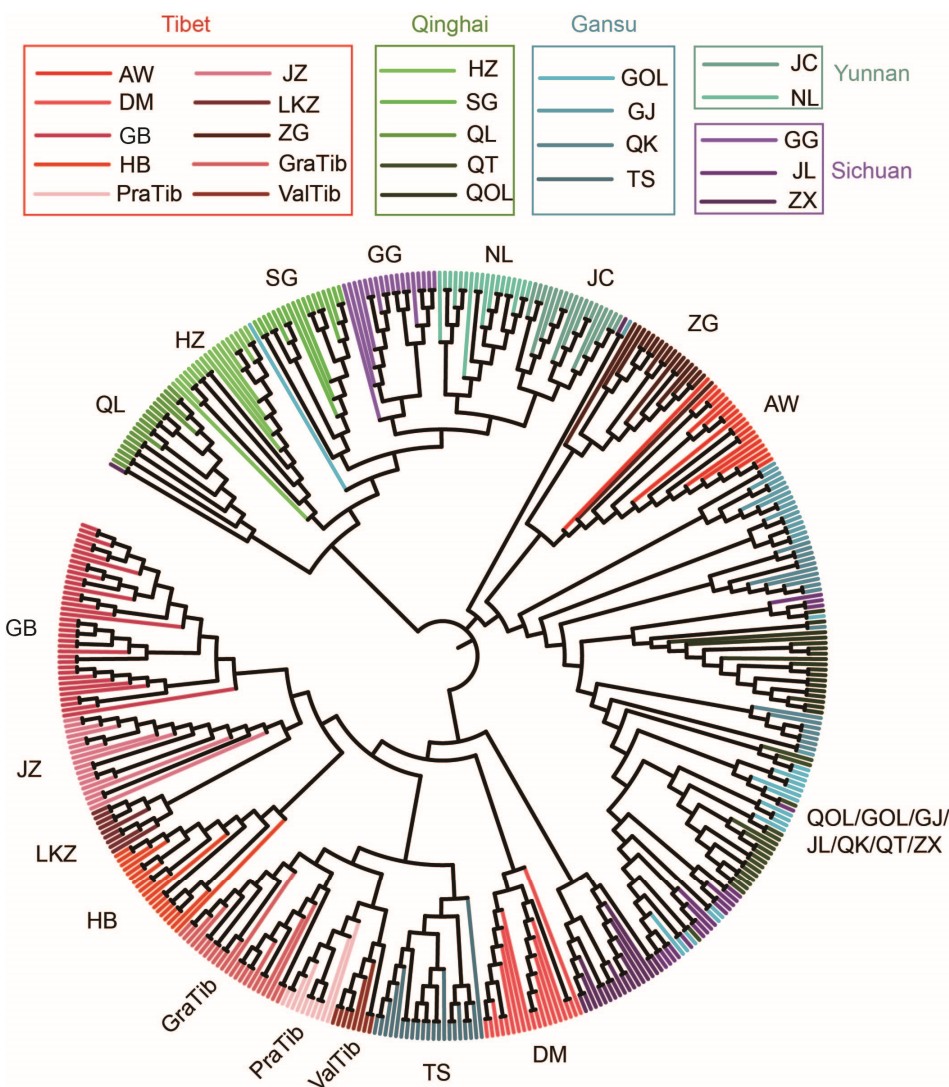

**Figure 2.** The NJ tree of 24 sheep breeds. The color of the symbols that indicate the geographic regions are the same as those in the PCA plots. The red color represents Tibetan sheep in Tibet (N = 180—AW, Awang sheep from Gongjue County in Tibet, 19; DM, Duoma sheep from Anduo County in Tibet, 19; GB, Gangba sheep from Gangba County in Tibet, 37; HB, Huoerba sheep from Zhongba County in Tibet, 19; JZ, Jiangzi sheep from Jiangzi County in Tibet, 18; LKZ, Langkazi sheep from Langkazi County in Tibet, 9; ZG, Zuogong sheep from Zuogong County in Tibet, 18; ValTib, Shangu sheep from Gongga County in Tibet, 8; PraTib, grassland sheep from Dangxiong County in Tibet, 10; GraTib, grassland sheep from Gaize County in Tibet, 23). The cyan color represents Tibetan sheep in Yunnan (N = 36—JC, Jianchuan sheep from Jianchuan County in Yunnan, 18; NL, Ninglang sheep from Ninglang Autonomous County in Yunnan, 18). The green color represents Tibetan sheep in Qinghai (N = 90—HZ, Heizang sheep from Guinan County in Qinghai, 18; SG, Shangu sheep from Maqin County in Qinghai, 18; QL, Qilian sheep from Qilian County in Qinghai, 18; QT, Qinghai Tibetan sheep from Zeku County in Qinghai, 18; QOL, Oula sheep from Henan County in Qinghai, 18). The blue color represents Tibetan sheep in Gansu (N = 75—GOL, Oula sheep from Maqu County in Gansu, 18; GJ, Ganjia sheep from Xiahe County in Gansu, 18; QK, Qiaoke sheep from Luqu County in Gansu, 18; TS, Tao sheep from Zhuoni County in Gansu, 21). The purple color represents Tibetan sheep in Sichuan (N = 55—GG, Gongga sheep from Luding County in Sichuan, 18; JL, Jialuo sheep from Aba County in Sichuan, 18; ZXK, Zhaxika sheep from Shiqu County in Sichuan, 19).

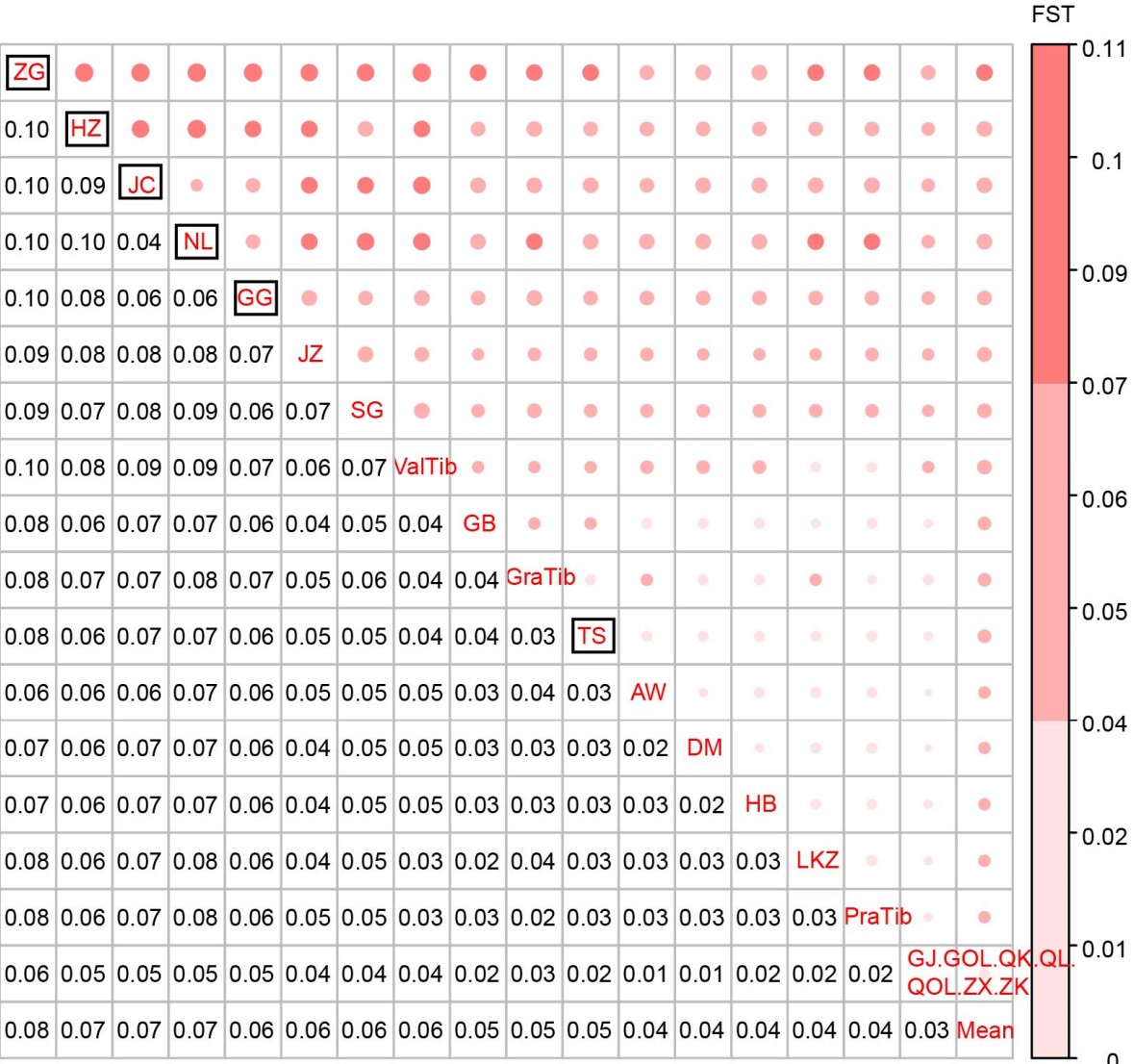

**Figure 3.** Population genetic differentiation distance (FST) between two Tibetan sheep populations. The values in the bottom-left represent population divergence (FST) between each of the two populations. The populations in the black boxes show medium population differentiation levels in five Tibetan regions, respectively.

Therefore, a breed-specific SNP set was successfully constructed for the Tibetan sheep population of Zuogong in Tibet, and a total of four sheep breeds, including Tibetan Zuogon sheep, were labeled based on the population differentiation index (FST). The four breeds were from different Tibetan areas, namely Tibetan Zuogong sheep (SNP = 30) (Figure 5D,E, Supplementary Table S2), Qinghai Black Tibetan sheep (HZ, SNP = 60) (Figures 6A,B and S1 and Table S3), Sichuan Gongga sheep (GG, SNP = 60) (Figures 6C,D and S2 and Table S4) and Gansu Tao sheep (TS, SNP = 30) (Figures 6E,F and S3 and Table S5). Finally, a total of 170 SNPs were obtained from these four Tibetan sheep populations for subsequent breed identification.

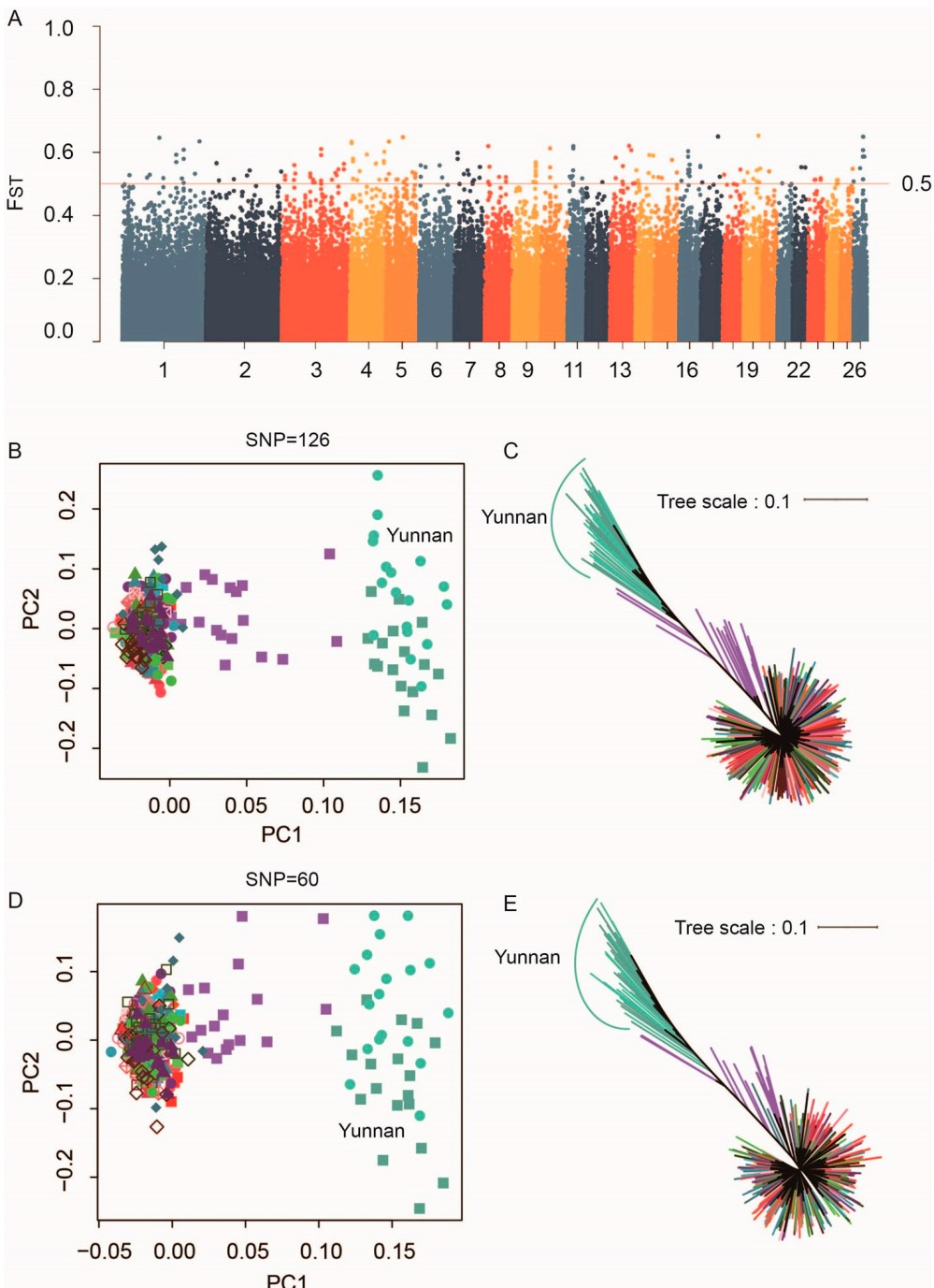

**Figure 4.** Construction of specific SNP genetic marker sets in Yunnan Tibetan sheep. (**A**) Genome-wide distribution of FST values in Yunnan Tibetan sheep (Jianchuan (JC) and Ninglang (NL) sheep). The significance threshold of selection signature was set to 0.5 for each individual test and is indicated with red horizontal full lines. (**B**) PCA and (**C**) NJ tree analysis of 126 SNPs distinguishing sheep from Yunnan (JC and NL). (**D**) PCA and (**E**) NJ tree analysis of 60 SNPs distinguishing sheep from Yunnan (JC, Jianchuan, and NL, Ninglang). The cyan solid square and circle represent JC and NL sheep in PCA plots, respectively. The color of JC and NL sheep in the NJ tree was the same as that in the PCA plots.

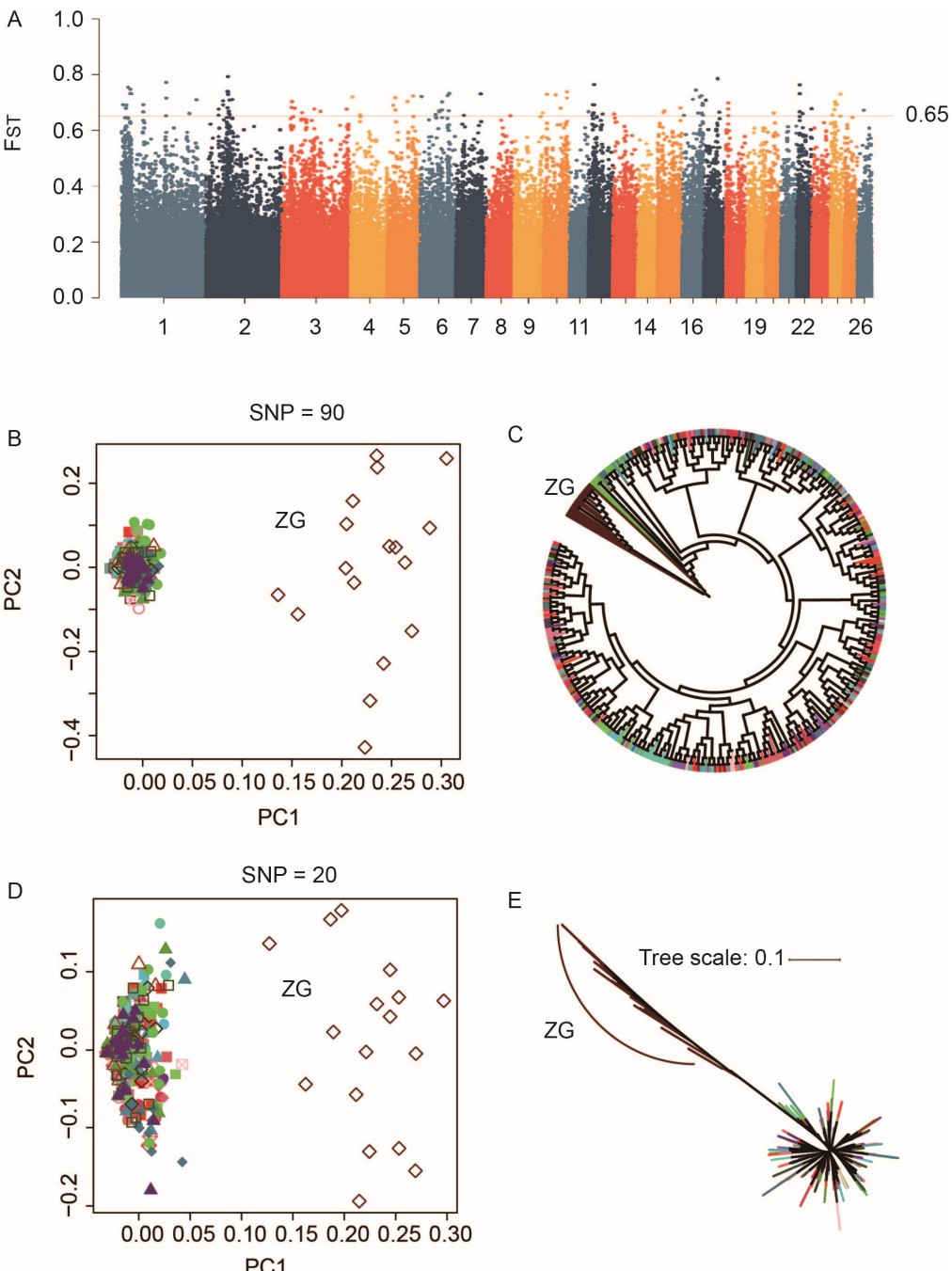

**Figure 5.** Construction of specific SNP genetic marker sets in Zuogong (ZG) Tibetan sheep.
(**A**) Genome-wide distribution of FST values in Yunnan–Tibetan sheep (Jianchuan, JC, and Ninglang,
NL, sheep). The significance threshold of the selection signature was set to 0.65 for each individual
test and is indicated with red horizontal full lines. (**B**) PCA and (**C**) NJ tree analysis of 90 SNPs
distinguishing Zuogong (ZG) sheep. (**D**) PCA and (**E**) NJ tree analysis of 20 SNPs distinguishing
Zuogong (ZG) sheep. The dark red hollow rhombus represents ZG sheep in the PCA plots. The color
of ZG sheep in the NJ tree was as the same as that in the PCA plots.

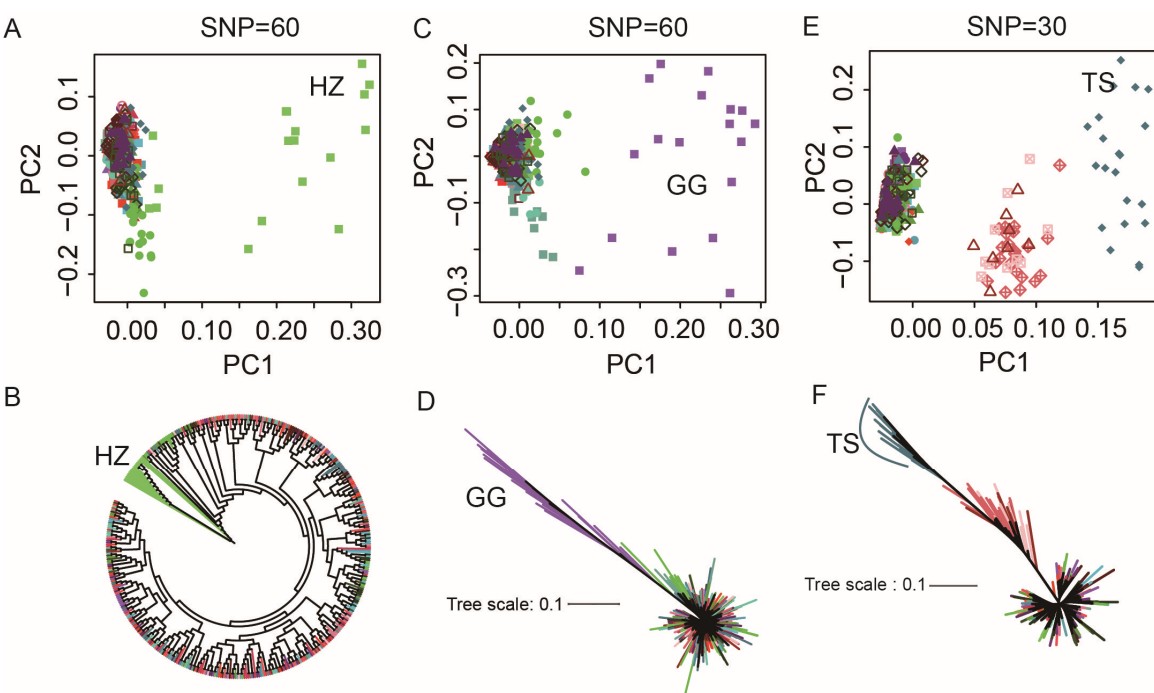

**Figure 6.** Construction of specific SNP genetic marker sets in Qinghai Heizang (HZ), Sichuan Gongga (GG), and Gansu Tao (TS) sheep. (**A**) PCA and (**B**) NJ tree analysis of 60 SNPs distinguishing sheep. (**C**) PCA and (**D**) NJ tree analysis of 60 SNPs distinguishing Gongga (GG) sheep. (**E**) PCA and (**F**) NJ tree analysis of 30 SNPs distinguishing Tao sheep (TS). The green solid square represents HZ sheep in PCA plots. The purple solid square represents Gongga sheep (GG) in PCA plots. The blue solid rhombus represents Tao sheep (TS) in PCA plots. The color of these three sheep breeds in the NJ tree were the same as that in the PCA plots.

## 4. Discussion

Molecular genetic markers (i.e., DNA molecular polymorphisms) have the characteristics of rich polymorphisms, genetic stability, and high accuracy, and have been widely used in genetic breeding, gene mapping, and species origin, evolution, and taxonomy. At present, commonly used DNA molecular markers mainly include the following: restriction fragment length polymorphisms (RFLP) [22]; random amplified polymorphic DNA (RAPD) [23]; amplified fragment length polymorphism (AFLP) [24], and single-nucleotide-sequence polymorphisms (SNPs). Among them, SNP markers have high density, wide distribution, low mutation frequency based on single nucleotides, high genetic stability, rapid detection, and simple analysis, and have become the mainstream of molecular genetic markers, with broad application prospects in genetic analysis [25]. The use of SNP markers for breed identification has been studied on cattle and horses [26–28]. However, the establishment of the characteristic SNP genetic signatures of Tibetan sheep breeds is insufficient.

In recent years, research on Tibetan sheep has mainly focused on their adaptive genes, such as hypoxia adaptation [16,29], wool growth [30], and the analysis of population genetic structure [31,32]. However, there are few reports on the construction of specific SNP marker sets for the identification of each Tibetan sheep population. Wei et al. [32] used Illumina Ovine SNP50 chip data to analyze the population structure of 10 local sheep breeds in China. The results showed that Mongolian sheep and Kazakh sheep in North China clustered together, while Tibetan sheep in Southwest China clustered alone. Based on this, the local sheep in China are divided into the following two types: one is the fat-tailed type, such as Mongolian sheep and Kazakh sheep, and the other is the thin-tailed type, such as Tibetan sheep. Yang et al. [2] performed whole-genome resequencing of 77 local Chinese sheep. Through population structure analysis and gene flow results, the Chinese local

sheep could be divided into three genetic groups (Qinghai–Tibet, Yunnan–Guizhou, and northern and eastern China), Consistently, our study found the Tibetan sheep in Yunnan region can also be distinguished from other Tibetan sheep populations, showing that these Tibetan sheep population have obvious regional specificity. Although such studies have contributed to improving our knowledge of the genetic basis and structure of Tibetan sheep, the identifying characteristic SNP genetic markers of Tibetan sheep breeds still remain unclear. In this study, based on 600k SNP genotype data combined with selection signal analysis strategy, we firstly constructed a Yunnan region-specific genetic marker set. At the same time, PCA and NJ tree results all showed that Tibetan Zuogong sheep (ZG), Qinghai Black Tibetan sheep (HZ), Sichuan Gongga sheep (GG), and Gansu Tao sheep (TS) have independent genetic backgrounds. These results are consistent with their geographic location. Based on this, we also constructed these four characteristic SNP breed-specific genetic marker sets. This makes up for the lack of the construction of Tibetan sheep breed identification tags, and at the same time provides a genome-level basis for the scientific classification and accurate identification of livestock and poultry genetic resources on the Qinghai–Tibet Plateau.

Additionally, in accordance with our earlier studies [2,3], the PCA and NJ tree based on the 600k SNP array data sets separated Yunnan Tibetan sheep and Tibet Tibetan sheep from all the Tibetan sheep populations. At the same time, it was found that there were great differences in population genetic differentiation distance between Yunnan-Tibetan sheep and other areas, which was consistent with the results of PCA and NJ trees and indicated that the Tibetan sheep in Yunnan have an independent genetic background. Based on this, we screened region-specific SNP loci with selection signal analysis, and then used PCA and NJ tree methods to confirm each other in order to construct a high-quality region-specific locus of Tibetan sheep in Yunnan. The number of SNP marker was 60 SNPs. The analysis found that this newly constructed region-specific marker set had 100% accuracy, and that it could also clearly distinguish Tibetan sheep in Yunnan from other regions, achieving the same discriminatory effect as the 600k chip data. This greatly reduces the cost of germplasm resource identification and greatly promotes the progress of local sheep breed identification in China. Additionally, the construction of Tibetan sheep SNP markers can not only provide a scientific basis for solving the "same name and different species" and "different names" of Tibetan sheep at the molecular level, but can also be applied to the identification of new genetic resources of Tibetan sheep and breed protection [15].

## 5. Conclusions

In conclusion, the genetic structure analysis and population genetic differentiation distance (FST) showed that Yunnan–Tibetan sheep (NL, Ninglang; JC, Jianchuan), Zuogong (ZG), Heizang (HZ), Gongga (GG), and Tao sheep (TS) can be clearly distinguished from other Tibetan sheep populations and had medium population differentiation levels (FST ≥0.05). Based on the population differentiation index FST, the PCA and NJ tree results showed that only 60 specific SNPs can successfully separate Tibetan sheep in Yunnan region from Tibetan sheep in other regions, and the distinguishing effect on Yunnan–Tibetan sheep reached 100%. Using the same method, we found that the four Tibetan sheep breeds, including Zuogong (ZG, 20 SNPs), Heizang (HZ, 60 SNPs), Gongga (GG, 60 SNPs) and Tao sheep (TS, 30 SNPs), can also be distinguished from other Tibetan sheep populations with only a few SNP loci (20–60), and the distinguishing effect reached 100%. Overall, we successfully obtained a Yunnan region-specific (60 SNPs) genetic marker set and four breed-specific SNP genetic marker sets (20–60 SNPs) for the first time for the identification of Tibetan sheep breeds at the molecular level. These makes up for the lack of the construction of Tibetan sheep breed identification tags, and at the same time provide a genome-level basis for scientific classification and accurate identification of livestock and poultry genetic resources on the Qinghai–Tibet Plateau.

**Supplementary Materials:** The following supporting information can be downloaded at: https://www.mdpi.com/article/10.3390/agriculture13020322/s1, Figure S1: Construction of specific SNP genetic marker sets in Qinghai Black Tibetan sheep (HZ). Figure S2: Construction of specific SNP genetic marker sets in Sichuan Gongga Tibetan sheep (GG). Figure S3: Construction of specific SNP genetic marker sets in Gansu Tao sheep (TS). Table S1: Population distribution based on 600K high density SNP arrays.; Table S1: Population distribution based on 600K high density SNP arrays. Table S2: Information on 20 breed-specific SNP genetic markers used to distinguish Zuogong sheep (ZG). Table S3: Information on 60 breed-specific SNP genetic markers used to distinguish Heizang sheep (HZ). Table S4: Information on 60 breed-specific SNP genetic markers used to distinguish Gongga sheep (GG). Table S5: Information on 30 breed-specific SNP genetic markers used to distinguish Tao sheep (TS). Table S6: Information on 60 breed-specific SNP genetic markers used to distinguish Tibetan sheep in Yunnan.

**Author Contributions:** Formal analysis, B.L. and Y.Z.; data curation, J.H. and X.H.; writing—original draft preparation, B.L. and Y.Z.; writing—review and editing, L.J. and J.L.; visualization, B.L.; resources, Y.P. and B.D.; project administration, L.J.; funding acquisition, L.J. and Y.M. All authors have read and agreed to the published version of the manuscript.

**Funding:** This project was supported by the National Natural Science Foundation of China (Nos. 32222079 and 31961143021); the Modern Wool Sheep Technology Research System (CARS-39-01); and the Science and Technology Innovation Project of the Chinese Academy of Agricultural Sciences (ASTIP-IAS01).

**Institutional Review Board Statement:** All experiments in this study involving animals were conducted according to the ethical policies and procedures approved by the Animal Care and Use Committee of the Chinese Academy of Agricultural Sciences and the Ministry of Agriculture of the People's Republic of China (IAS2019-57).

**Data Availability Statement:** The data that support the findings of this study have been deposited into the CNGB Sequence Archive (CNSA) [33] of China National GeneBank DataBase (CNGBdb) [34] with accession number CNP0003687.

**Acknowledgments:** Thanks to researchers from Tibetan areas in the Tibet, Qinghai, Gansu, Sichuan, and Yunnan provinces for their help in collecting Tibetan sheep samples.

**Conflicts of Interest:** The authors declare no conflict of interest.

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
