# Peer review of "Establishment of Tibetan-Sheep-Specific SNP Genetic Markers"

_agriculture, doi:10.3390/agriculture13020322_

Round 1

Reviewer 1 Report

The present investigation is about "SNP genetic markers for the identification of Tibetan sheep breeds". This research provides interesting information. However, it is necessary to make some changes before its final publication.

MATERIAL AND METHODS

General comments: I suggest adding in the "Sample collection" section in which the characteristics of the animals are mentioned, such as average ages, etc. In addition, what were the criteria for the selection of these animals.

Specific comments:

Line 113-134.- In the "Sample collection" section, I suggest to summarize or remove this part because this is already mentioned in "Supplementary Table S1. Population distribution based on 600K high density SNP arrays.” Also, review the format of this section "Justified"

RESULT

Specific comments:

Line 253.- Correct the word "Figure 1".

Line 296-297.- I suggest changing this sentence to the “Materials and Methods” section, “FST values, named fixation index, are typically used to evaluate differentiated genomic regions and identify selective signals among whole-genomic sequences [21].” as well as describe it a little more

CONCLUSIONS

Line 476.- I suggest making a more specific conclusion. Also, remove a parenthesis “(TS))”

Author Response

The present investigation is about "SNP genetic markers for the identification of Tibetan sheep breeds". This research provides interesting information. However, it is necessary to make some changes before its final publication.

MATERIAL AND METHODS

General comments: I suggest adding in the "Sample collection" section in which the characteristics of the animals are mentioned, such as average ages, etc. In addition, what were the criteria for the selection of these animals.

Response: Thank you for your good suggestions. All experimental sheep were unrelated adult sheep. In addition, all experimental sheep were randomly selected.

Specific comments:

Line 113-134.- In the "Sample collection" section, I suggest to summarize or remove this part because this is already mentioned in "Supplementary Table S1. Population distribution based on 600K high density SNP arrays.” Also, review the format of this section "Justified"

Response: Thank you for your good suggestions. According to your suggestion, we summarized the “Sample collection” section and removed unnecessary information. Also, we modified the format of this section.

Line 116-124. The sentence “In this study, a total of 448 individuals from 24 Tibetan sheep populations in five provinces, including 192 samples from ten Tibet–Tibetan sheep breeds, 90 samples from five Qinghai–Tibetan sheep breeds, 75 samples from four Gansu–Tibetan sheep breeds, 55 samples from three Sichuan–Tibetan sheep breeds and 36 samples from two Yunnan–Tibetan sheep breeds, were investigated (Supplementary Table S1). The 192 Tibet–Tibetan sheep samples were from ten populations, comprising 19 Awang sheep from Gongjue County (AW), 19 Duoma sheep from Anduo County (DW), 37 Gangba sheep from Gangba County (GB), 19 Huoerba sheep from Zhongba County (HB), 18 Jiangzi sheep from Jiangzi County (JZ), 19 Langkazi sheep from Langkazi County (LKZ), 18 Zuogong sheep from Zuogong County (ZG), 10 Shangu sheep from Gongga County(ValTib), 10 Grassland sheep from Dangxiong County (PraTib) and 23 Grassland sheep from Gaize County (GraTib). The 90 Qinghai–Tibetan sheep samples were from five breeds, comprising 18 Heizang sheep from Guinan County (HZ), 18 Shangu sheep from Maqin County (SG), 18 Qilian sheep from Qilian County (QL), 18 Qinghai Tibetan sheep from Zeku County (QT) and 18 Oula sheep from Henan County (QOL). The 75 Gansu–Tibetan sheep samples from four breeds comprised 18 Oula sheep from Maqu County (GOL), 18 Ganjia sheep from Xiahe County (GJ), 18 Qiaoke sheep from Luqu County (QK) and 21 Tao sheep from Zhuoni County (TS). The 55 Sichuan–Tibetan sheep samples from three breeds comprised 18 Gongga sheep from Luding County (GG), 18 Jialuo sheep from Aba County (JL), 19 Zhaxika sheep from Shiqu County (ZXK). The 36 samples from two Yunnan–Tibetan sheep breeds comprised 18 Jianchuan sheep from Jianchuan County (JC) and 18 Ninglang sheep from Ninglang Autonomous County (NL). The basic information and phenotypic data of individuals were included in the phenotypic database of livestock and poultry genetic resources on the Qinghai–Tibet Plateau, and all tissue samples were used for subsequent DNA extraction and SNP BeadChip genotyping assay.”has been modified to “In this study, a total of 448 unrelated adult individuals from 24 Tibetan sheep populations in five provinces, including 192 samples from ten Tibet–Tibetan sheep breeds, 90 samples from five Qinghai–Tibetan sheep breeds, 75 samples from four Gansu–Tibetan sheep breeds, 55 samples from three Sichuan–Tibetan sheep breeds and 36 samples from two Yunnan–Tibetan sheep breeds, were investigated (Supplementary Table S1). All experimental sheep were randomly selected. The basic information and phenotypic data of individuals were included in the phenotypic database of livestock and poultry genetic resources on the Qinghai–Tibet Plateau, and all tissue samples were used for subsequent DNA extraction and SNP BeadChip genotyping assay”

RESULT

Specific comments:

Line 253.- Correct the word "Figure 1".

Response: Thank you for your comments. We have changed it in line 247 as “Figure 1”

Line 296-297.- I suggest changing this sentence to the “Materials and Methods” section, “FST values, named fixation index, are typically used to evaluate differentiated genomic regions and identify selective signals among whole-genomic sequences [21].” as well as describe it a little more.

Response: Thank you for your good suggestions. According to your suggestion, we changed the sentence to the “Materials and Methods” section, “FST values, named fixation index, are typically used to evaluate differentiated genomic regions and identify selective signals among whole-genomic sequences [21].” as well as describe it a little more.

Line 145-150, we added the sentence “FST values, named fixation index, are typically used to evaluate differentiated genomic regions and identify selective signals among whole-genomic sequences. The population differentiation index FST is an important indicator to detect whether there is genetic differentiation between populations, and the degree of differentiation between populations is reflected through the difference in allele or genotype frequency between populations ”

CONCLUSIONS

Line 476.- I suggest making a more specific conclusion. Also, remove a parenthesis “(TS))”

Response: Thank you for your good suggestions. According to your suggestion, In line 535-551, the sentence “In conclusion, we firstly constructed a region- specific and four breed-specific SNP genetic marker sets, which can distinguish Tibetan sheep populations in Yunnan region from other Tibetan sheep populations with only 60 SNP loci and the four Tibetan sheep populations (Tibetan Zuogong sheep (ZG), Qinghai Black Tibetan sheep (HZ), Sichuan Gongga sheep (GG) and Gansu Tao sheep (TS) can also be distinguished from other Tibetan sheep populations with only a few SNP loci (20-60). This makes up for the lack of the construction of Tibetan sheep breed identification tags, and at the same time provide a genome-level basis for scientific classification and accurate identification of livestock and poultry genetic resources on the Qinghai–Tibet Plateau.” has been modified to “In conclusion, the genetic structure analysis and population genetic differentiation distance(FST)showed Yunnan–Tibetan sheep(NL, Ninglang; JC, Jianchuan), Zuogong (ZG), Heizang (HZ), Gongga (GG,) and Tao sheep (TS) can be clearly distinguished from other Tibetan sheep populations and had medium population differentiation levels (FST ≥0.05). Based on the population differentiation index FST, the PCA and NJ tree results showed with only 60 specific SNPs can successfully separate Tibetan sheep in Yunnan region from Tibetan sheep in other regions and the distinguishing effect on Yunnan–Tibetan sheep reached 100%. Using the same method, we found that the four Tibetan sheep breed, including Zuogong (ZG, 20 SNPs), Heizang (HZ, 60 SNPs), Gongga (GG, 60 SNPs) and Tao sheep (TS, 30 SNPs), can also be distinguished from other Tibetan sheep populations with only a few SNP loci (20-60) and the distinguishing effect reached 100%. Overall, we succesfully obtained a Yunnan region- specific (60 SNPs) genetic marker sets and four breed-specific SNP genetic marker sets (20-60 SNPs) for the first time for the identification of Tibetan sheep breeds at the molecular level. These makes up for the lack of the construction of Tibetan sheep breed identification tags, and at the same time provide a genome-level basis for scientific classification and accurate identification of livestock and poultry genetic resources on the Qinghai–Tibet Plateau.”

Reviewer 2 Report

The authors for the first time Establishment of Tibetan-sheep-specific SNP genetic markers.

The purpose and objectives are fulfilled by the authors in full. 

However, despite this, some remarks are made.  These comments are noted in the text of the manuscript.

The remarks made do not diminish the scientific significance of the manuscript material!

The study is aimed at solving an actual scientific problem. Namely, the establishment of Tibetan-sheep-specific SNP genetic markers. Tibetan sheep are the largest livestock in the Qinghai-Tibet Plateau. They play an important role in the economy, culture and society of the Tibetan pastoral areas. Due to long-term natural selection, the complexity of the highland environment has created a variety of breeds of Tibetan sheep, which is mainly reflected in their morphology, physiology and genetics. Therefore, studies on the resource assessment of Tibetan sheep are of practical importance for their genetic selection, rational protection and use. Experimental studies were carried out by the authors taking into account the requirements of ethical policies and procedures approved by the Committee for the Care and Use of Animals of the Chinese Academy of Agricultural Sciences and the Ministry of Agriculture of the People's Republic of China (IAS2019-57). In the study, the material was subjected to a sufficient volume with the involvement of modern equipment and materials! The material of the manuscript is written in accessible and understandable English. The authors correctly use professional and special terminology and vocabulary. Based on the results of studying the experimental material, the authors draw objective conclusions and conclusions. References to literature are justified and do not infringe the copyrights of third parties. Strengths of the manuscript. The authors managed to conduct large-scale and large-scale studies at a high scientific and methodological level. The presented material reflects the totality of SNP genetic markers specific to the Tibetan sheep population. This topic is poorly disclosed in the scientific community. The research design is formulated according to the subject of the manuscript and accurately characterizes the essence of the problem raised. The authors comprehensively and well-reasoned give explanations and answers to established patterns and facts! Weaknesses of the manuscript. The structure of the manuscript for individual positions does not meet generally accepted scientific requirements. For example, a manuscript abstract does not contain a research goal. The results of the study are described superficially and vaguely! In the Introduction section, the purpose and objectives of this study are also missing. Some drawings require technical editing. With this in mind, I believe that the manuscript can be printed after minor revision. Recommendations to authors for editing the manuscript. 1. Make changes to the abstract of the manuscript. That is, add the purpose of the study, clarify the results obtained (write in more detail). 2. In the Introduction, add the purpose and objectives of the study. 3. Supplement the drawings of the manuscript with the signatures of the reflected materials. Those. add the names of the axes, their units of measurement. On some figures, remove unnecessary captions (for example, Figures 4 and 5). Check the titles of the figures (for example, figure 3). 4. In conclusion, write about the results obtained (specifically). Do not use generalized concepts and expressions!

All comments are noted in the text of the manuscript. Look in the document attached to the letter!

Author Response

The authors for the first time Establishment of Tibetan-sheep-specific SNP genetic markers.

The purpose and objectives are fulfilled by the authors in full. 

However, despite this, some remarks are made.  These comments are noted in the text of the manuscript.

The remarks made do not diminish the scientific significance of the manuscript material!

The study is aimed at solving an actual scientific problem. Namely, the establishment of Tibetan-sheep-specific SNP genetic markers. Tibetan sheep are the largest livestock in the Qinghai-Tibet Plateau. They play an important role in the economy, culture and society of the Tibetan pastoral areas. Due to long-term natural selection, the complexity of the highland environment has created a variety of breeds of Tibetan sheep, which is mainly reflected in their morphology, physiology and genetics. Therefore, studies on the resource assessment of Tibetan sheep are of practical importance for their genetic selection, rational protection and use. Experimental studies were carried out by the authors taking into account the requirements of ethical policies and procedures approved by the Committee for the Care and Use of Animals of the Chinese Academy of Agricultural Sciences and the Ministry of Agriculture of the People's Republic of China (IAS2019-57). In the study, the material was subjected to a sufficient volume with the involvement of modern equipment and materials! The material of the manuscript is written in accessible and understandable English. The authors correctly use professional and special terminology and vocabulary. Based on the results of studying the experimental material, the authors draw objective conclusions and conclusions. References to literature are justified and do not infringe the copyrights of third parties. Strengths of the manuscript. The authors managed to conduct large-scale and large-scale studies at a high scientific and methodological level. The presented material reflects the totality of SNP genetic markers specific to the Tibetan sheep population. This topic is poorly disclosed in the scientific community. The research design is formulated according to the subject of the manuscript and accurately characterizes the essence of the problem raised. The authors comprehensively and well-reasoned give explanations and answers to established patterns and facts! Weaknesses of the manuscript. The structure of the manuscript for individual positions does not meet generally accepted scientific requirements. For example, a manuscript abstract does not contain a research goal. The results of the study are described superficially and vaguely! In the Introduction section, the purpose and objectives of this study are also missing. Some drawings require technical editing. With this in mind, I believe that the manuscript can be printed after minor revision. Recommendations to authors for editing the manuscript.

  1. Make changes to the abstract of the manuscript. That is, add the purpose of the study, clarify the results obtained (write in more detail).

Response: Thank you for your comments. According to your comments, we made changes to the abstract of the manuscript and added the purpose of study and rewrote the results in more detail.

Line 18-37. The sentence “Tibetan sheep are one of the three major coarse sheep breeds in China and they possess a long history of formation. However, few studies have been conducted on the construction of specific single-nucleoside polymorphism (SNP) genetic marker sets for the identification of Tibetan sheep breeds at the molecular level. In this study, a total of 448 individuals from 24 Tibetan sheep populations in five regions of Tibet, Qinghai, Gansu, Yunnan and Sichuan were analyzed using the Affymetrix Ovine SNP 600K high-density chip. We further analyzed the genetic structure and the population differentiation index (FST) of these Tibetan sheep populations. Firstly, a region-specific SNP genetic marker set of Yunnan–Tibetan sheep was successfully constructed according to the regional division; the set comprised 60 specific SNPs, and the principle component analysis (PCA) and neighbor-joining (NJ) tree results showed that the distinguishing effect of PCA and NJ tree on Yunnan–Tibetan sheep was basically the same, and the discrimination effect reached 100%. Using the same method, we then successfully constructed four breed-specific SNP marker sets, including Zuogong (ZG, 20), Heizang (HZ, 60), Gongga (GG, 60) and Tao sheep (TS, 30). In conclusion, we have constructed a region- specific and four breed-specific SNP genetic marker sets for the first time, which can distinguish Tibetan sheep populations in Yunnan from other Tibetan sheep populations with only 60 SNP loci and the four Tibetan sheep populations can also be distinguished from other Tibetan sheep populations with only a few SNP loci (20-60). These made up for the lack of genetic marker sets for the identification of Tibetan sheep breeds, and provided a genomic basis for the scientific classification and accurate identification of livestock and poultry genetic resources on the Qinghai–Tibet Plateau.” has been modified to “Tibetan sheep are one of the three major coarse sheep breeds in China and they possess a long history of formation. However, few studies have been conducted on the identification of Tibetan sheep breeds at the molecular level. In this study, a total of 448 individuals from 24 Tibetan sheep populations in five regions of Tibet, Qinghai, Gansu, Yunnan and Sichuan were analyzed using the Affymetrix Ovine SNP 600K high-density chip to construct specific single-nucleoside polymorphism (SNP) genetic marker sets of Tibetan sheep breeds. Firstly, the genetic structure analysis showed Yunnan–Tibetan sheep (NL, Ninglang; JC, Jianchuan), Zuogong (ZG), Heizang (HZ), Gongga (GG,) and Tao sheep (TS) can be clearly distinguished from other Tibetan sheep populations. Next, based on the population differentiation index FST, the PCA and NJ tree results showed with only 60 specific SNPs can successfully separate Tibetan sheep in Yunnan region from Tibetan sheep in other regions and the distinguishing effect on Yunnan–Tibetan sheep reached 100%. Using the same method, we found that the four Tibetan sheep breed, including Zuogong (ZG, 20 SNPs), Heizang (HZ, 60 SNPs), Gongga (GG, 60 SNPs) and Tao sheep (TS, 30 SNPs), can also be distinguished from other Tibetan sheep populations with only a few SNP loci (20-60) and the distinguishing effect reached 100%. Overall, we successfully obtained a Yunnan region- specific (60 SNPs) genetic marker sets and four breed-specific SNP genetic marker sets (20-60 SNPs) for the first time for the identification of Tibetan sheep breeds at the molecular level. These made up for the lack of genetic marker sets for the identification of Tibetan sheep breeds, and provided a genomic basis for the scientific classification and accurate identification of livestock and poultry genetic resources on the Qinghai–Tibet Plateau.” 

  1. In the Introduction, add the purpose and objectives of the study.

Response: Thank you for your comments. According to your comments, we added the purpose and objectives of the study in the introduction.

Line 97-98. we added the sentence “In order to construct specific single-nucleoside polymorphism (SNP) genetic marker sets for the identification of Tibetan sheep breeds at the molecular level.”

  1. Supplement the drawings of the manuscript with the signatures of the reflected materials. Those. add the names of the axes, their units of measurement. On some figures, remove unnecessary captions (for example, Figures 4 and 5). Check the titles of the figures (for example, figure 3).

Response: Thank you for your comments. In the figure 3, The genetic differentiation index (FST) is a ratio and has no unit of measurement. The values in the bottom-left represented  genetic differentiation index (FST) between each two populations. We added the names of the axes in the top-right.

On Figures 4 and 5, unnecessary captions was removed.

And we have checked the titles of the all figures.

  1. In conclusion, write about the results obtained (specifically). Do not use generalized concepts and expressions!

Response: Thank you for your comments. According to your comments, we rewrote the results specifically in conclusion.

Line 535-551, the sentence “In conclusion, we firstly constructed a region- specific and four breed-specific SNP genetic marker sets, which can distinguish Tibetan sheep populations in Yunnan region from other Tibetan sheep populations with only 60 SNP loci and the four Tibetan sheep populations (Tibetan Zuogong sheep (ZG), Qinghai Black Tibetan sheep (HZ), Sichuan Gongga sheep (GG) and Gansu Tao sheep (TS) can also be distinguished from other Tibetan sheep populations with only a few SNP loci (20-60). This makes up for the lack of the construction of Tibetan sheep breed identification tags, and at the same time provide a genome-level basis for scientific classification and accurate identification of livestock and poultry genetic resources on the Qinghai–Tibet Plateau.” has been modified to “In conclusion, the genetic structure analysis and population genetic differentiation distance(FST)showed Yunnan–Tibetan sheep(NL, Ninglang; JC, Jianchuan), Zuogong (ZG), Heizang (HZ), Gongga (GG,) and Tao sheep (TS) can be clearly distinguished from other Tibetan sheep populations and had medium population differentiation levels (FST ≥0.05). Based on the population differentiation index FST, the PCA and NJ tree results showed with only 60 specific SNPs can successfully separate Tibetan sheep in Yunnan region from Tibetan sheep in other regions and the distinguishing effect on Yunnan–Tibetan sheep reached 100%. Using the same method, we found that the four Tibetan sheep breed, including Zuogong (ZG, 20 SNPs), Heizang (HZ, 60 SNPs), Gongga (GG, 60 SNPs) and Tao sheep (TS, 30 SNPs), can also be distinguished from other Tibetan sheep populations with only a few SNP loci (20-60) and the distinguishing effect reached 100%. Overall, we succesfully obtained a Yunnan region- specific (60 SNPs) genetic marker sets and four breed-specific SNP genetic marker sets (20-60 SNPs) for the first time for the identification of Tibetan sheep breeds at the molecular level. These makes up for the lack of the construction of Tibetan sheep breed identification tags, and at the same time provide a genome-level basis for scientific classification and accurate identification of livestock and poultry genetic resources on the Qinghai–Tibet Plateau.”

Reviewer 3 Report

The manuscript entitled: Establishment of Tibetan-sheep-specific SNP genetic markers, focuses on an important topic in sheep breeding, which finally constructed SNP marker sets that could discriminate four sheep breed-specific genetic markers, including 29 Zuogong (ZG, 20), Heizang (HZ, 60), Gongga (GG, 60) and Tao sheep (TS, 30). Indeed, the idea of the manuscript is novel and it is well written using the standard journal format.I recommend the acceptance of manuscript entitled: Establishment of Tibetan-sheep-specific SNP genetic markers.  The evaluation is based on that overall value of data presented and novelty of the idea. In addition, it is on the scope of this outstanding journal.

Author Response

Dear Reviewer:

Thank you very much for giving us comments on our manuscript.